# Interleukin-1 Receptor Antagonist Modulates Liver Inflammation and Fibrosis in Mice in a Model-Dependent Manner

**DOI:** 10.3390/ijms20061295

**Published:** 2019-03-14

**Authors:** Raphael P. H. Meier, Jeremy Meyer, Elisa Montanari, Stephanie Lacotte, Alexandre Balaphas, Yannick D. Muller, Sophie Clément, Francesco Negro, Christian Toso, Philippe Morel, Leo H. Buhler

**Affiliations:** 1Visceral and Transplant Surgery, Department of Surgery, Geneva University Hospitals and Medical School, 1211 Geneva, Switzerland; Jeremy.Meyer@hcuge.ch (J.M.); Elisa.Montanari@unige.ch (E.M.); Stephanie.Lacotte@unige.ch (S.L.); Alexandre.Balaphas@hcuge.ch (A.B.); Christian.Toso@hcuge.ch (C.T.); Philippe.Morel@hcuge.ch (P.M.); Leo.Buhler@hcuge.ch (L.H.B.); 2Transplant Surgery, University of California San Francisco, San Francisco, CA 94143, USA; Yannick.Muller@ucsf.edu; 3Division of Clinical Pathology, Geneva University Hospitals and Medical School, 1211 Geneva, Switzerland; Sophie.Clement@unige.ch (S.C.); Francesco.Negro@hcuge.ch (F.N.); 4Division of Gastroenterology and Hepatology, Geneva University Hospitals and Medical School, 1211 Geneva, Switzerland

**Keywords:** liver fibrosis, interleukin-1 receptor antagonist, interleukin-1, bile duct ligation, carbon tetracholoride, insulin

## Abstract

Background: Interleukin-1 (IL-1)β and IL-1 receptor antagonist (IL-1Ra) have been proposed as important mediators during chronic liver diseases. We aimed to determine whether the modulation of IL-1β signaling with IL-1Ra impacts on liver fibrosis. Methods: We assessed the effects of IL-1β on human hepatic stellate cells (HSC) and in mouse models of liver fibrosis induced by bile duct ligation (BDL) or carbon tetrachloride treatment (CCl-4). Results: Human HSCs treated with IL-1β had increased IL-1β, IL-1Ra, and MMP-9 expressions in vitro. HSCs treated with IL-1β had reduced α-smooth muscle actin expression. These effects were all prevented by IL-1Ra treatment. In the BDL model, liver fibrosis and Kuppfer cell numbers were increased in IL-1Ra KO mice compared to wild type mice and wild type mice treated with IL-1Ra. In contrast, after CCl-4 treatment, fibrosis, HSC and Kupffer cell numbers were decreased in IL-1Ra KO mice compared to the other groups. IL-1Ra treatment provided a modest protective effect in the BDL model and was pro-fibrotic in the CCl-4 model. Conclusions: We demonstrated bivalent effects of IL-1Ra during liver fibrosis in mice. IL-1Ra was detrimental in the CCl-4 model, whereas it was protective in the BDL model. Altogether these data suggest that blocking IL-1-mediated inflammation may be beneficial only in selective liver fibrotic disease.

## 1. Introduction

Interleukin-1 (IL-1) is a pro-inflammatory cytokine playing a key role in acute and chronic inflammation [1,2,3,4]. IL-1 refers to two similar cytokines which bind to the same receptor, one being mostly secreted (IL-1β) and the other remaining intracellular (IL-1α) [2]. The inflammasome promotes the maturation of IL-1β in response to infectious or non-infectious agents. IL-1β is produced by stimulated monocytes, macrophages and to a lesser extent by other cell types, such as neutrophils, epithelial and endothelial cells, smooth muscle cells and fibroblasts [5]. IL-1β activity is regulated by IL-1 receptor antagonist (IL-1Ra), a naturally anti-inflammatory cytokine that binds to type 1 IL-1 receptors and that is produced by activated myeloid cells, hepatocytes, fibroblasts [2] and or mesenchymal stem cells (MSCs) [6,7]. IL-1β has been proposed as an important mediator of inflammation and tissue damage in chronic liver disease [8]. Indeed, elevated circulating levels of both IL-1β and IL-1Ra are detected in the serum of patients with chronic liver diseases, including alcoholic liver disease [9], chronic hepatitis B and C, and primary biliary cirrhosis [10,11,12]. More recently, IL-1β was shown to be implicated in the transformation of steatosis to steatohepatitis and liver fibrosis [13,14]. However, determining whether IL-1β is a contributor or a consequence to liver fibrosis pathogenesis demands further investigation. In this context, two functional polymorphisms in the IL-1 gene cluster were identified as related to development of acute-on-chronic liver failure [15]. Blocking IL-1 pathway during liver injury using IL-1Ra represents an interesting strategy to determine its role in liver diseases. We previously observed that IL-1Ra knockout (KO) mice had delayed liver regeneration after partial hepatectomy [16], which was consistent with observations from others [17,18]. More recently, we demonstrated that 23% to 36% of MSC subpopulations release IL-1Ra in vitro and in vivo, and that MSC-secreted cytokines show an anti-fibrotic effect in mice [19]. This MSC-IL-1Ra-mediated hepatoprotective effect was further confirmed by others in acute liver injury models [20,21]. The important role of IL-1β signaling in the progression from chronic liver injury to fibrosis was previously characterized in mice [8] and rats [22]. However, the mechanisms involving IL-1Ra and its potential protecting effect in hepatic fibrogenesis remain to be elucidated.

In the present study, we aimed to investigate whether IL-1Ra is regulating, along with IL-1β, the progression of chronic liver injury to fibrosis. In vitro, we showed that IL-1β upregulated itself and IL-1Ra in human stellate cells (HSC); IL-1Ra prevented these effects. In vivo, we observed that IL-1β and IL-1Ra were strongly upregulated in mice liver after bile duct ligation (BDL) or carbon tetrachloride treatment (CCl-4) and further showed that IL-1Ra KO mice displayed increased fibrosis in the BDL model and reduced fibrosis in the CCl-4 model. We thus highlighted a dual effect of IL-1Ra that was dependent on the type of liver injury.

## 2. Results

### 2.1. Interleukin-1 Receptor Antagonist and Liver Fibrosis

#### 2.1.1. IL-1β Reduce Stellate Cell Activation In Vitro

α-smooth muscle actin (α-SMA) expression by activated HSCs is a key feature of liver fibrosis [23]. Therefore, we investigated the effect of IL-1β on α-SMA mRNA and protein expressions in human HSCs in vitro, using immunoblotting and RT-PCR. Treatment of HSCs with IL-1β modestly decreased α-SMA protein expression levels in primary HSCs when compared to control (Figure 1A,B). This effect was prevented when IL-1Ra was added. Transforming growth factor-β1 (TGF-β1) (used as a positive control) induced increased α-SMA mRNA expression under these conditions (Figure 1C). Of note, IL-1Ra was not able to reverse the effect of TGF-β1 (Appendix A), suggesting that the IL-1Ra effect is specific to IL-1β induction. We then sought to analyze whether IL-1β and IL-1Ra are regulated in HSCs, the key cell in the transition from inflammation to fibrosis [24]. IL-1β and IL-1Ra mRNA expressions were significantly upregulated in HSCs after exogenous IL-1β treatment (by 3.2- and 1.4-fold, respectively) (Figure 1D,E). Treatment with IL-1Ra prevented IL-1β effects and further reduced IL-1β and IL-1Ra expressions below basal levels. No effect on IL-1β and IL-1Ra expression was observed following HSC treatment with TGF-β (Figure 1D,E). Matrix metalloproteinases expression was previously demonstrated to be regulated by IL-1α in HSCs [25,26]. Therefore, we analyzed whether this regulatory effect persisted when using IL-1β. MMP-9 mRNA expression level was upregulated in IL-1β-treated human HSCs (by 2.1-fold) (Figure 1F). This effect was prevented when cells were treated with IL-1Ra. Again, TGF-β had no effect. MMP-2 and collagen type I expression was not significantly modified following IL-1β and L-1Ra treatment (Figure 1G,H). Overall, these results indicated that IL-1β reduces stellate cell activation and enhances IL-1β, IL-1Ra and MMP-9 production in vitro.

#### 2.1.2. IL-1β and IL-1Ra Expression Levels are Upregulated in Mice after BDL or CCl-4-Induced Liver Fibrosis

We analyzed liver IL-1β and IL-1Ra mRNA expression levels using RT-PCR, in both BDL- and CCl-4-induced liver fibrosis models. In both models, liver IL-1β mRNA expression levels were significantly upregulated in all mice groups following liver fibrosis induction (Figure 2A,B). IL-1β expression was the highest in either WT mice (BDL model) or in IL-1Ra KO mice (CCl-4 model). IL-1β expression was the lowest in IL-1Ra treated mice in both models. As expected, IL-1Ra expression was absent in IL-1Ra KO mice in both fibrosis models (Figure 2C,D). In line with gene expression findings, corresponding elevations of serum IL-1β and IL-1Ra were found in the BDL model (Figure 2E,F) (not done in the CCl-4 model). Comparison between WT and KO baseline IL-1β and IL-1Ra mRNA levels in sham animals without fibrosis induction is provided in Appendix A. Overall, these results showed that IL-1β and IL-1Ra are upregulated during BDL-or CCl-4-induced liver fibrosis.

#### 2.1.3. IL-1Ra Has Bivalent Effects on Liver Fibrosis in BDL and CCl-4-Induced Liver Fibrosis

To investigate the contributing role of IL-1Ra in hepatic fibrogenesis, we quantified liver fibrosis in WT, IL-1Ra KO and IL-1Ra treated mice over time after BDL or CCl-4 injection. Quantification of collagen on sirius red stained sections showed that hepatic fibrosis was significantly increased in IL-1Ra KO mice following BDL and decreased in IL-1Ra KO mice following CCl-4 injections (Figure 3A–D). IL-1Ra treatment in WT mice caused decreased fibrosis in the BDL model and increased fibrosis in the CCl-4 model. Of note, IL-1Ra KO mice were rescued following IL-1Ra treatment (Appendix A). Quantification of liver mRNA expression of Collagen type 1α (Coll1α) confirmed these results (Figure 3E,F). Comparison between WT and KO baseline Coll1α mRNA levels in sham animals without fibrosis induction is provided in Appendix A. Altogether, these results demonstrated that the effects of IL-1Ra knockout and treatment during liver fibrosis are model specific. While IL-1Ra deficiency increases liver fibrosis in the BDL model, it decreases liver fibrosis in the CCl-4 model.

#### 2.1.4. Hepatic Stellate Cell Activation

To investigate whether the increased liver fibrosis induced by BDL or CCl-4 was paralleled by HSC activation, we determined α-SMA expression using immunofluorescence and RT-PCR on livers from the three groups (Figure 4). Quantification of α-SMA by immunofluorescence on liver sections and RT-PCR in liver showed that α-SMA expression remained globally unchanged in the BDL group (Figure 4A–C). In the CCl-4 model, α-SMA varied along with fibrosis intensity. WT mice and WT mice treated with IL-1Ra had significantly more α-SMA positive cells and liver α-SMA mRNA expression compared to IL-1Ra KO mice (Figure 4D–F). Comparison between WT and KO baseline α-SMA mRNA levels in sham animals without fibrosis induction is provided in Appendix A. These results confirm the model-specific variations in liver fibrosis intensity following IL-1Ra Knockout or treatment.

#### 2.1.5. Kupffer Cell Activation 

We then investigated the activation of Kupffer cells during liver fibrosis in the two models by quantifying the ionized calcium binding adaptor molecule 1 (IBA-1) [27,28] on liver sections (Figure 5). The results varied along with fibrosis intensity in each model, respectively. In the BDL model, the numbers of liver IBA-1 expressing cells were significantly increased in IL-1Ra KO mice compared to WT mice or IL-1Ra treated mice (Figure 5A,B). The opposite situation was observed in the CCl-4 model; a lower level of activation was observed in the IL-1Ra KO group compared to the two other groups (Figure 5C,D). Overall, these results are consistent with those observed in fibrosis level variations and further highlight model specific effects during IL-1Ra modulation.

#### 2.1.6. Liver Enzymes

Hepatocyte death and liver injury were assessed by measuring alanine transaminase (ALT) in mice serum in the two groups (Figure 6). In the BDL model, WT mice had higher level of ALT compared to IL-1Ra KO mice or IL-1Ra treated mice (Figure 6A). In the CCl-4 model, WT mice treated with IL-1Ra had higher ALT levels compared to WT mice or IL-1Ra KO mice (Figure 6B). IL-1Ra KO mice had the lowest ALT level in the later model.

#### 2.1.7. Matrix Metalloproteinases

We further analyzed MMP-2, MMP-9, MMP-13 and tissue inhibitor of matrix metalloproteinases 1 (TIMP-1) hepatic gene expressions after BDL- and CCl-4-induced liver fibrosis using RT-PCR (Figure 7). In the BDL model, MMP-2, 9,13 and TIMP-1 levels were all higher in the WT mice group compared to the IL-1Ra KO mice group or WT mice treated with IL-1Ra (Figure 7A–D). In the CCl-4 model, MMP-2 and MMP-13 were higher in the IL-1Ra KO mice group compared to the two other groups (Figure 7E,G). Regarding MMP-9 and TIMP-1, expression was decreased in the IL-1Ra KO mice group compared to the two other groups (Figure 7F,H). Comparison between WT and KO baseline α-SMA mRNA levels in sham animals without fibrosis induction is provided in Appendix A. These results confirm that IL-1Ra regulates genes in a model specific-manner during liver fibrosis.

## 3. Discussion

Acute and chronic liver injuries are accompanied by a major inflammatory response, including an increased expression of IL-1β and its natural antagonist IL-1Ra [8,10,11,12,16,22,29,30,31,32,33,34,35,36,37,38,39]. We previously demonstrated that the absence of IL-1Ra caused an increased inflammatory state and a delayed liver regeneration in a mouse model of partial hepatectomy [16] and that IL-1Ra might be an important factor in MSC antifibrotic effect in the liver [19].

In the present study, we analyzed the effect of IL-1β alone or blocked with IL-1Ra on primary human HSCs, as well as the effect of IL-1Ra deficiency or supplementation in two models of in vivo experimental liver fibrosis. Our in vitro experiments showed that the treatment of primary human HSCs with IL-1β resulted in an increase of its own transcription but reduced α-SMA expression, with the effects being prevented by recombinant IL-1Ra. In vivo, IL-1β and IL-1Ra expressions were strongly upregulated in WT mice livers following BDL and CCl-4 injuries, confirming the implication of IL-1 signaling in this process and its regulation by the liver [40,41]. The knockout of IL-1Ra resulted in opposite patterns in the two liver fibrosis models; IL-1Ra KO mice had increased fibrosis in the BDL model while fibrosis was reduced in the CCl-4 model. It is known that BDL ligation causes a periportal fibrosis [42] while CCl-4 intoxication leads to an initial centrilobular matrix deposition [43]. Considering our in vitro results and the known predominance of HSCs in the pericentral area [44], we hypothesized that IL-1β signaling modulates liver fibrosis in an HSC-dependent manner.

A first question that we aimed to address was to understand the effect of increased IL-1 signaling on HSC, the key player in liver fibrosis. We knew from previous works that HSCs respond to IL-1 stimulation in vitro with various effects on proliferation and α-SMA expression [24,26,45,46]. We made the hypothesis that IL-1β, which is a key cytokine during inflammation, may increase HSC activation. However, our results indicate that, on the contrary, upon IL-1β treatment, HSC have a reduced activation. These findings are consistent with those reported previously, where HSC activation was shown to be inhibited after treatment with IL-1α [45] or IL-1β [24]. As suggested previously, IL-1β might mobilize and increase proliferation of HSC rather than directly promoting fibrosis production [24]. Knowing that HSCs are the major source of proteases that degrade basement membrane collagens such as MMP-9 [25], we analyzed its expression in vitro. We found that IL-1β strongly increased MMP-9 gene transcription in HSCs and this effect was prevented by IL-1Ra. These findings are consistent with those of others showing that, under 3D HSC cell culture conditions, stimulation with IL-1α caused robust induction of pro-MMP-9 [25,26]. Of note, enhanced HSC activation was observed in the latter experiments, however IL-1α was used instead of IL-1β and this may explain the difference with our findings. These results add further evidence that IL-1β is a key regulator of MMP-9 expression by HSC during liver fibrosis [8,25,26]. In addition, we [19] and others [47] previously demonstrated that liver MMP-9 expression increases via mesenchymal stem cells’ paracrine effect and might be a protective factor during liver fibrosis. Overall, whether it is clear from numerous previous works that prolonged inflammation can lead to fibrosis [48], our findings along with those of others [24] suggest that this process is not solely driven by a direct effect of IL-1 on HSC.

We then thought to investigate the effect of IL-1 signaling downregulation on HSC activation and fibrosis in vivo using IL-1Ra KO mice. In both models, IL-1β was upregulated upon liver fibrosis induction in the liver (2–5-fold) and serum (4–6-fold). IL-1Ra was strongly upregulated in the liver (15–50-fold) and serum (35-fold); knowing that its main production source is the liver [49]. The increase in IL-1 signaling (in IL-1Ra KO mice) resulted in increased fibrosis in the BDL model reduced fibrosis in the CCl-4 model. These results were consistent within their respective models and corresponding changes in collagen type I mRNA expression and Kupffer cells activation were observed. We hypothesized that these conflicting results may represent a model-specific influence. Using IL-1 receptor knockout mice, Pradere et al. found a non-significant decrease in liver fibrosis in both BDL and CCl-4 models [24]. They further showed that antagonizing both IL-1 and TNF-α was essential to significantly reduce NF-κB in HSCs and reduce liver fibrosis in a BDL model (no data in the CCl-4 model). In their experiments, IL-1 and TNF-α did not lead to HSC activation but promoted survival of activated HSCs. Consistent with the work of Pradere et al., Gieling et al. found that IL-1 receptor-deficient mice exhibited reduced fibrosis after thioacetamide treatment for 8 weeks [8]. Overall, these data are in accordance with what we observed in the BDL model (more fibrosis in IL-1Ra KO mice and diminished HSC activation). Altogether, it is likely that both IL-1 and TNF-α are both required to successfully translate inflammation into fibrosis; however, increased IL-1 signaling by itself may contribute to an increase in liver fibrosis. Regarding IL-1 signaling in the CCl-4 model, previous evidence is scarce and difficult to compare with our experiments. Based on our consistent findings in the IL-1Ra KO group showing diminished fibrosis, collagen type I expression, activated HSC numbers, α-SMA and TIMP-1 expression, we can only hypothesize a direct effect of CCl-4-mediated centrilobular increased inflammation and IL-1 signaling on pericentral HSC. Given our observation that IL-1 diminish HSC activation in vitro, we hypothesize that when inflammation is mainly pericentral (CCl-4 model), IL-1Ra absence (namely, unopposed IL-1 signaling) ultimately partially block the profibrogenic signals. All together our results suggest that blocking IL-1-mediated inflammation may be beneficial mostly in biliary type liver fibrosis. Certain IL-1β gene polymorphisms has been associated with liver fibrosis progression in primary biliary cholangitis [50,51] and further confirm this eventuality. Drug-related injury to the liver may then not be amenable to benefit from IL-1β signaling inhibition since the injury is predominantly centrilobuar. Finally, since IL-1β is implicated in virus clearance, viral hepatitis does not represent good candidates for IL-1β inhibition [52,53].

In our experiments, we also investigated the effect of IL-1Ra treatment (anakinra) and observed that it provided marginal protective effect on liver fibrosis after BDL and increased fibrosis in the CCl-4 model. Using a different model and animal species, Mancini et al. observed a decreased collagen content and fewer activated HSC in WT rats subjected to dimethylnitrosamine-induced liver fibrosis and treated with IL-1Ra [22]; we could marginally reproduce these findings in the BDL model only. Our primary hypothesis to explain this modest improvement is a failure of exogenous IL-1Ra to efficiently increase circulating levels due a possible negative feed-back mechanism regulating its own expression (Figure 2F). Indeed, we observed that IL-1Ra blood levels were not much increased compared to the WT group despite daily anakinra administration. Another argument in favor of this hypothesis is that IL-1Ra expression by the liver was reduced upon IL-1Ra administration. In the clinical setting, IL-1Ra treatment was demonstrated to be successful in situations in which IL-1Ra is absent or nonfunctional [54] while a very limited success was observed with recombinant IL-1Ra to treat human inflammatory disease [2] suggesting that endogenous IL-1Ra production cannot be surpassed. Of note, in a confirmatory experiment, we could demonstrate that IL-1Ra treatment can rescue IL-1Ra KO mice. However, it is also possible that IL-1Ra given systemically is not fully equivalent to the endogenous natural section of IL-1Ra. An example of that discrepancy is depicted by the difference observed in fasting insulin levels between the different groups of mice after BDL. We observed a significant increase in fasting insulin levels in mice after BDL in WT mice and KO mice (consistent with insulin resistance that is commonly observed during liver fibrosis/dysfunction [55]), however that increase was completely reversed by IL-1Ra treatment in WT mice (and KO mice) (Appendix A), consistently with the known anti-diabetic effect of exogenous IL-1Ra [56]. In the CCl-4 model, the profibrogenic effect of IL-1Ra treatment was consistent with all the other findings in this model and further reinforces the hypothesis of a dual and topographical-dependent effect of IL-1 signaling modulation.

## 4. Materials and Methods

### 4.1. HSC Isolation and Culture

Human HSCs were obtained from biopsies of healthy liver parenchyma from three different patients undergoing partial hepatectomy. The protocol was approved by the local research ethics committee of the Department of Surgery of the Geneva University Hospital. All donors provided their written informed consent for use of the samples in the present study. HSCs were isolated as previously described [57,58]. Cells were cultured in 24-well plates (100,000 cells/well) in IMDM medium (Invitrogen, Basel, Switzerland) containing 10% FCS, penicillin, and streptomycin (Invitrogen, Basel, Switzerland) at 37 °C with 5% CO_2_. Cells were used for experiments between passages 3 to 6. HSCs were either left untreated, or treated with recombinant IL-1β (10 ng/mL, Thermo Fisher Scientific, Waltham, MA, USA) for 48 h, or IL-1β (10 ng/mL) and IL-1Ra (15 μg/mL, Kineret, Amgen Europe B.V, Breda, The Netherlands) together, or Transforming growth factor-β 1 (TGF-β1) (50 ng/mL, PreproTech, UK) for 48h. For α-SMA protein detection, HSCs were treated and cultured for 5 days.

### 4.2. Electrophoretic and Immunoblot Analysis

Proteins from lysed cells were separated on polyacrylamide gels (SDS-PAGE) and transferred to polyvinylidene difluoride membranes (Milipore, Billerica, MA, USA). Membranes were blocked with 5% skim milk in wash buffer (20 mM Tris–HCl, pH 7.4, 140 mM NaCl, 0.1% Tween 20) and incubated with anti-alpha smooth muscle actin (α-SMA) [59] (obtained from C. Chaponnier, Geneva, Switzerland) or anti-vimentin (Dako, Baar, Switzerland) antibodies diluted in blocking solution. Following three washes, membranes were incubated with peroxidase-conjugated goat anti-mouse antibodies (Molecular Probes Inc, Eugene, OR, USA) diluted 1:6000 in wash buffer. Proteins were revealed by chemiluminescence (ECL, Interchim Inc., Montluçon, France) on photographic film (GE healthcare, Chicago, IL, USA) and signals were quantified using the Quantity One software (PDI, Inc., Huntington Station, NY, USA) and normalized by the expression of GAPDH or vimentin.

### 4.3. Animals

DBA-1 mice were purchased from Janvier (Le Genest-St-Isles, France). IL-1Ra knockout (IL-1Ra KO) mice were generated from DBA-1 background and were a kind gift from Prof. Cem Gabay of Geneva University [60,61]. Eight to ten-week-old male mice were used. Liver fibrosis was induced by 2 to 4 weeks of BDL or 6 weeks of CCl-4 treatment, as described here below. All animal studies were approved by the Animal Ethics Committee of the Geneva Veterinarian Office and Geneva University, Geneva, Switzerland (protocol 1043/3603/2, Oct 1, 2010).

### 4.4. Fibrosis Induction in Mice

Liver fibrosis was induced by BDL or CCl-4. BDL: briefly, mice were anesthetized with isoflurane, a midline laparotomy was performed, and the common bile duct was dissected and cut between four ligatures under a dissecting microscope. CCl-4: Three ml per kilogram of CCl-4 50% (*v*/*v*) solution in corn oil (Sigma Co., Milan, Italy), containing 1.0 ml/kg of CCl-4, was administered by intraperitoneal injections twice a week for 6 weeks. All mice were maintained under standard conditions at the animal facility of Geneva University. Water and food were provided ad libitum. IL-1Ra treated animals received 50mg/kg/day recombinant IL-1Ra, intraperitoneally, every day from BDL (or CCl-4) to sacrifice (Anakinra/Kineret, Amgen Europe B.V, Breda, The Netherlands). Sham operated and sham injected (NaCl 0.9%) mice were used as controls. In accordance with the 3R principles of animal experimentation in our institution (reduce, refine, replace), the severity of the BDL model in DBA-1 mice prompted us to sacrifice mice as soon as they presented signs of suffering. Accordingly, mice were sacrificed between 2 to 4 weeks. No mortality was observed in the CCl-4 model and all animals were sacrificed at 6 weeks. Blood and liver samples were collected to be analyzed.

### 4.5. Assessment of Hepatic Fibrosis

Liver collagen content was determined using sirius red histochemistry as previously described [62]. Tissue sections were observed using an Axiophot microscope (Carl Zeiss AG, Feldbach, Switzerland) and images were acquired with an Axiocam color camera (Zeiss, Feldbach, Switzerland). Hepatic fibrosis extent was determined using morphometric quantification (MetaMorph Software, Universal Imaging, West Chester, PA, USA). 

### 4.6. Immunostaining for Alpha-Smooth Muscle Actin (α-SMA) and Ionized Calcium Binding Adaptor Molecule 1 (IBA-1)

Paraffin sections were dewaxed, and rehydrated using xylene/ethanol baths and then heated at 95 °C in a 10 mM/pH 6.0 sodium citrate bath for 10 min. The detection of activated HSCs was performed using anti α-SMA antibody (a gift from Christine Chaponnier, Geneva University) (1:50). The detection of macrophages was performed using an anti IBA-1 antibody (Wako Chemicals, Richmond, VA, USA) (1:500). The liver sections were incubated overnight at 4 °C with primary antibody diluted in PBS containing 0.1% BSA, washed in PBS, and incubated for 1 h with a secondary antibody (Alexa Fluor 488 goat-anti mouse) diluted (1:1000) in PBS containing 0.1% BSA. All sections were stained with 0.09% Evans blue solution. Images of immunostained sections were acquired using Axiophot microscope and Axiocam color camera. The percentage of positive cells for α-SMA or IBA-1 was determined using MetaMorph, Image J and Definiens Software. Cells area positive for α-SMA or IBA-1 staining were counted and normalized to the total liver surface. An average of 3000 cells were counted on 4 histology fields per animal.

### 4.7. Real-Time Polymerase Chain Reaction (RT-PCR)

RT-PCR was used to determine the expression levels of fibrosis-related genes. Total RNA was extracted from liver samples or cultured cells using Qiagen RNeasy Mini kit (Qiagen, San Diego, CA, USA) according to manufacturer’s instructions. cDNA was synthesized from 1 μg of total RNA using SuperScript III reverse transcriptase (Invitrogen, Basel, Switzerland). RT-PCR was performed using SYBR Green PCR Master Mix (Applied Biosystems Inc, San Diego, CA, USA), with 2 ng cDNA and 300 nM of each primer following the following protocol: two minutes at 50 °C, 10 min at 95 °C, and for 45 cycles of 15 s at 95 °C, and 60 s at 60 °C using a SDS 7900 HT machine (Applied Biosystems Inc.). Reactions were performed in three replicates on 384-well plate. Raw *C*_t_ values obtained with SDS 2.2 (Applied Biosystems Inc.) were imported in Excel software (Microsoft, Redmond, WA, USA) and normalization factor and fold changes were calculated using the GeNorm method [63]. Briefly, GeNorm was used to determine which normalization genes are the most stable in the sample population studied and select them for normalization. We computed a normalization factor for each individual sample represented by the geometric mean of the quantity values from the most stable normalization genes. Each sample will therefore have its own normalization factor. Primers used for amplification were designed using Primer3 online software (http://frodo.wi.mit.edu/) unless otherwise specified (Table 1). All primers were tested with Amplifix Software (http://ifrjr.nord.univ-mrs.fr/AmplifX-Home-page) and blasted on http://www.ensembl.org. Human matrix metalloproteinase (MMP) -2 was purchased from Qiagen, San Diego, CA, USA. The efficiency of each primer was tested using positive control cDNA serial dilutions and negative control.

### 4.8. Serum Assays

Serum ALT was measured using UniCel DxC 800 Synchron Clinical Systems (Beckman Coulter, Inc., Brea, CA, USA), following manufacturer’s instructions. Serum IL-1β, IL-1Ra and fasting insulin contents were measured using mouse IL-1β ELISA set (BD Bioscience, San Jose, CA, USA), mouse IL-1Ra ELISA kit (RayBio, Norcross GA, USA) respectively, and Ultrasensitive Mouse Insulin ELISA (Mercodia, Uppsala, Sweeden) following manufacturer’s instructions.

### 4.9. Statistical Analysis

Results were expressed as means ± SEM. Differences between groups were analyzed using the Student *t*-test or Mann-Whitney U Test and one-way analysis of variance with Kruskal-Wallis testing corrections. A *p*-value < 0.05 was considered to be statistically significant.

## 5. Conclusions

In conclusion, our study demonstrated that IL-1Ra has opposite effects in two different liver fibrosis models. IL-1Ra was detrimental in the CCl-4 model, whereas it was protective in the BDL model. Since IL-1β can decrease α-SMA expression in primary HSC, and given the predominant pericentral distribution of HSC and pericentral inflammation in the CCl-4 model, we postulated that this may represents the cause of decreased fibrosis in the latter model. Altogether these data suggest that blocking IL-1-mediated inflammation with IL-1Ra may only be beneficial in selective liver fibrotic disease.

## Figures and Tables

**Figure 1 ijms-20-01295-f001:**
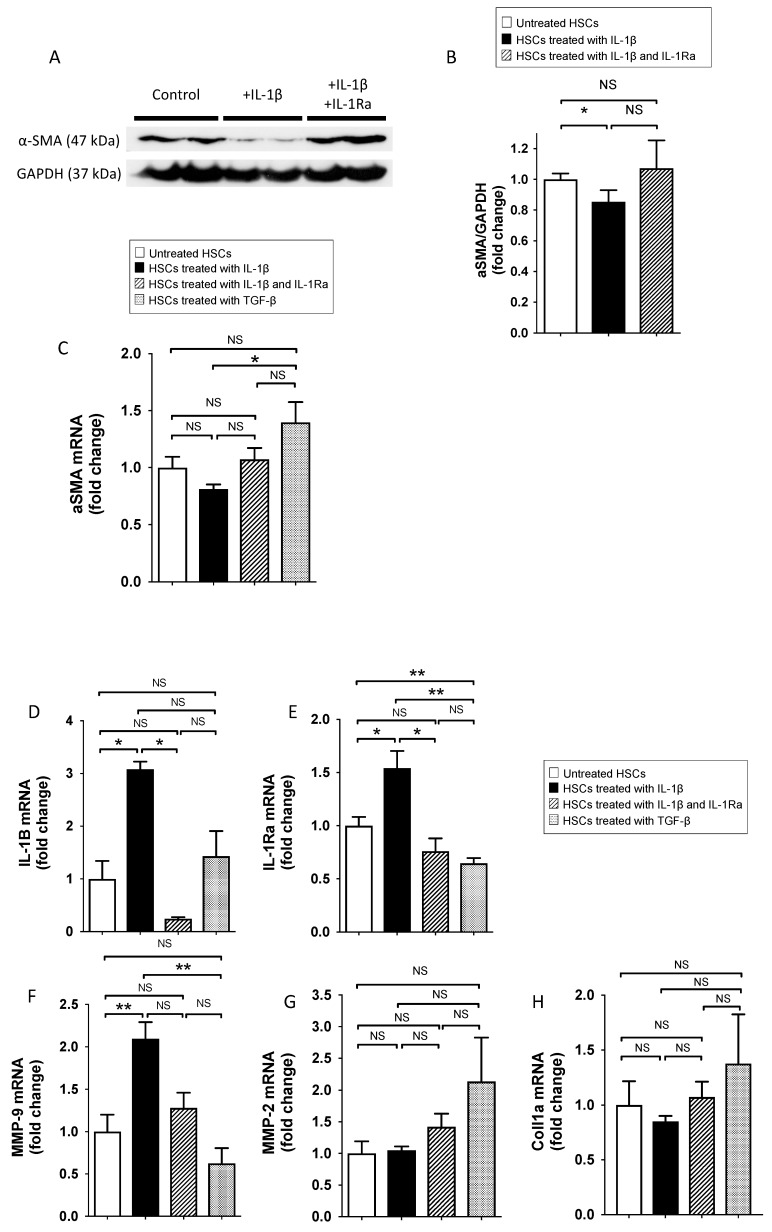
IL-1β reduces human stellate cell activation and increases the expression of IL-1β, IL-1Ra and matrix metalloproteinase (MMP) 9 in vitro. (**A**) Total protein extracts from human untreated HSCs (control), HSCs treated with IL-1β, and HSCs treated with IL-1β and IL-1Ra simultaneously were obtained. Protein extracts were subjected to SDS-PAGE, transferred to nitrocellulose and blotted with anti-α-SMA and GAPDH. (**B**) α-SMA signals were quantified by densitometry and normalized using GAPDH signals as a loading control. α-SMA (**C**), IL-1β (**D**), IL-1Ra (**E**), MMP-2 (**F**), MMP-9 (**G**), and collagen type 1 (**H**) mRNA expression levels were measured by RT-PCR. Data are mean values obtained from four independent experiments. NS p > 0.05, * *p* < 0.05, ** *p* < 0.01, comparing two groups as indicated.

**Figure 2 ijms-20-01295-f002:**
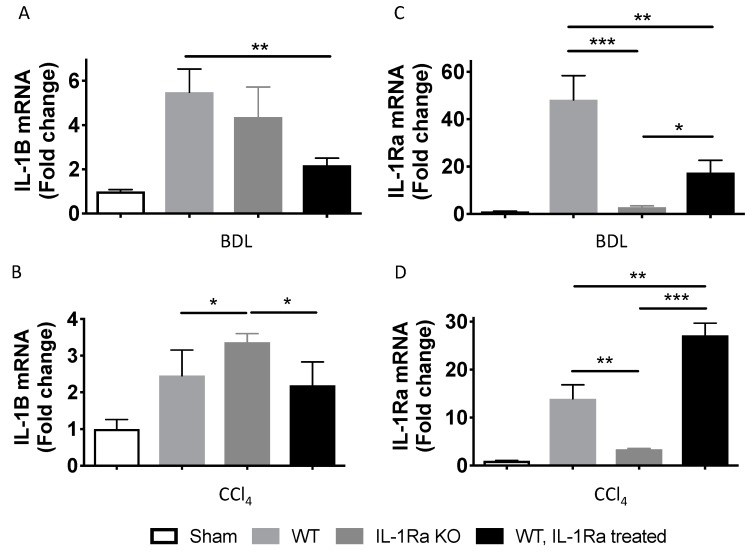
IL-1β and IL-1Ra expression levels are upregulated in mice after bile duct ligation- or carbon tetrachloride-induced liver fibrosis. IL-1β (**A**,**B**) and IL-1Ra (**C**,**D**) liver mRNA levels were measured by RT-PCR in control mice and mice following 2–4 weeks BDL or 6 weeks CCl-4 treatment. BDL and CCl-4 groups were respectively as follows: WT (*n* = 7 and *n* = 8), IL-1Ra KO (*n* = 7 and *n* = 4) and IL-1Ra treated mice (50 mg/kg/day) (*n* = 10 and *n* = 13). Sham treated mice were used as control for the BDL and CCl-4 groups (*n* = 8 and *n* = 4). Serum levels of IL-1β (**E**) and IL-1Ra (**F**) were measured by ELISA. * *p* < 0.05, ** *p* < 0.01, *** *p* < 0.001, comparing two groups as indicated.

**Figure 3 ijms-20-01295-f003:**
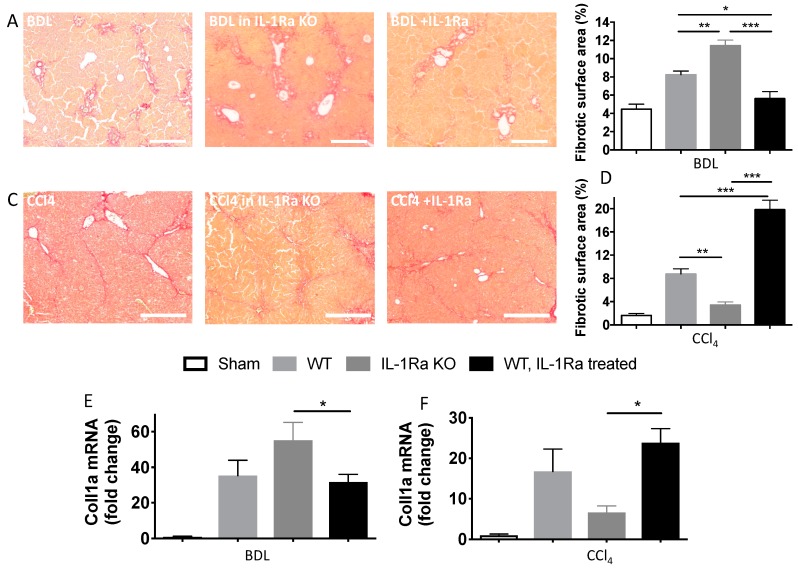
Differential and model-dependent variation of liver fibrosis following IL-1β signaling modulation with IL-1Ra knockout versus treatment. Mice livers were fixed in formalin and embedded in paraffin and collected for mRNA extraction and RT-PCR following 2–4 weeks BDL or 6 weeks CCl-4 treatment. BDL and CCl-4 groups were respectively as follows: WT (*n* = 7 and *n* = 8), IL-1Ra KO (*n* = 7 and *n* = 4) and IL-1Ra treated mice (50 mg/kg/day) (*n* = 10 and *n* = 13). Sham treated mice were used as control for the BDL and CCl-4 groups (*n* = 8 and *n* = 4, histology not shown). Liver sections stained by sirius red (**A**). On sirius staining liver parenchyma appears in light yellow and fibrotic areas appear in red. Morphometric quantification of fibrosis was performed on multiple liver sections and expressed as percentage of liver surface area for the BDL (**B**) and CCl-4 groups (**D**). Liver collagen type I mRNA levels were measured by RT-PCR in mice with BDL (**E**) or CCl-4 treatment (**F**). Scale bars, 400 µm. * *p* < 0.05, ** *p* < 0.01, *** *p* < 0.001, comparing two groups as indicated.

**Figure 4 ijms-20-01295-f004:**
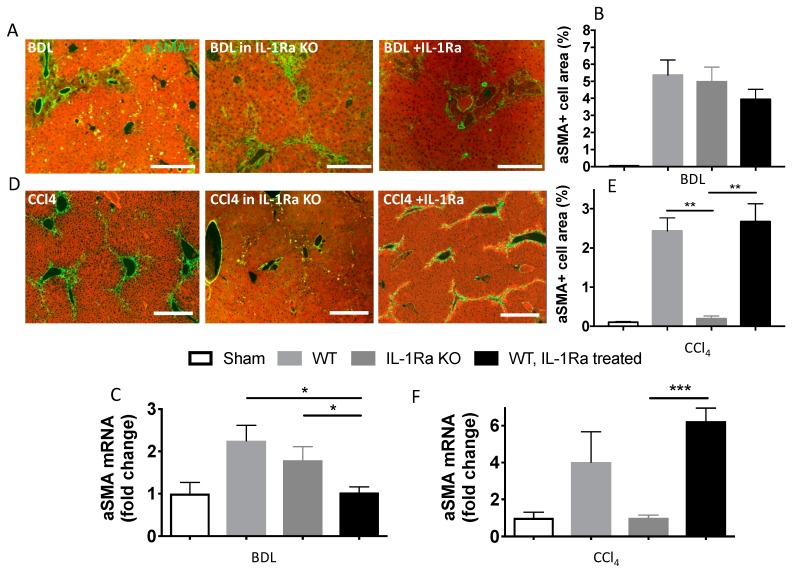
Differential and model-dependent variation in periportal hepatic stellate cell number and alpha-smooth muscle actin expression in fibrotic mice following IL-1Ra knockout or treatment. (A-F) Mice livers were fixed in formalin and embedded in paraffin and collected for mRNA extraction and RT-PCR following 2–4 weeks BDL or 6 weeks CCl-4 treatment. BDL and CCl-4 groups were respectively as follows: WT (*n* = 7 and *n* = 8), IL-1Ra KO (*n* = 7 and *n* = 4) and IL-1Ra treated mice (50 mg/kg/day) (*n* = 10 and *n* = 13). Sham treated mice were used as control for the BDL and CCl-4 groups (*n* = 8 and *n* = 4, histology not shown). (**A**,**D**) Liver sections immunostained for α-SMA (green) and stained with evans blue (red). Activated hepatic stellate cells appear in green and hepatic parenchyma areas appear in red. Morphometric quantification of stellate cell activation was performed on multiple liver sections and expressed as percentage of α-SMA+ area for the BDL (**B**) and CCl-4 (**E**) groups. Liver α-SMA mRNA levels were measured by RT-PCR in mice following BDL (**C**) or CCl-4 treatment (**F**). Scale bars, 400 µm. * *p* < 0.05, ** *p* < 0.01, *** *p* < 0.001, comparing two groups as indicated.

**Figure 5 ijms-20-01295-f005:**
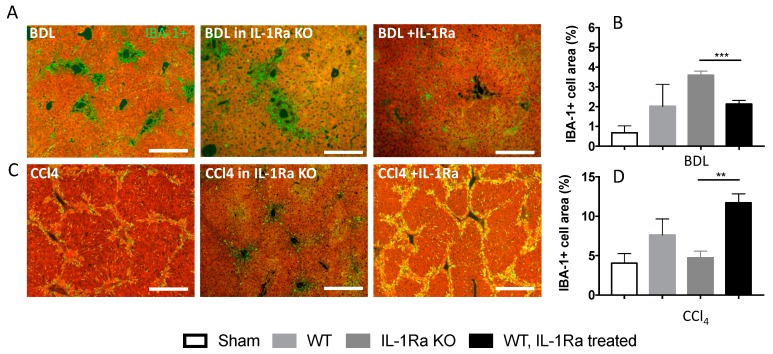
Differential and model-dependent variation in Kupffer cell number in fibrotic mice following IL-1Ra knockout or treatment. (**A**–**D**) Mice livers were fixed in formalin and embedded in paraffin following 2–4 weeks BDL or 6 weeks CCl-4 treatment. BDL and CCl-4 groups were respectively as follows: WT (*n* = 7 and *n* = 8), IL-1Ra KO (*n* = 7 and *n* = 5) and IL-1Ra treated mice (50 mg/kg/day) (*n* = 8 and *n* = 14). Sham treated mice were used as control for the BDL and CCl-4 groups (*n* = 8 and *n* = 6, histology not shown). (**A**,**C**) Liver sections immunostained for IBA-1 (green) and stained with evans blue (red). Activated Kupffer cells appear in green and hepatic parenchyma areas appear in red. Morphometric quantification of activated Kupffer cell numbers was performed on multiple liver sections and expressed as percentage of IBA-1 + area for the BDL (**B**) and CCl-4 (**D**) groups. Scale bars, 400 µm. ** *p* < 0.01, *** *p* < 0.001, comparing two groups as indicated.

**Figure 6 ijms-20-01295-f006:**
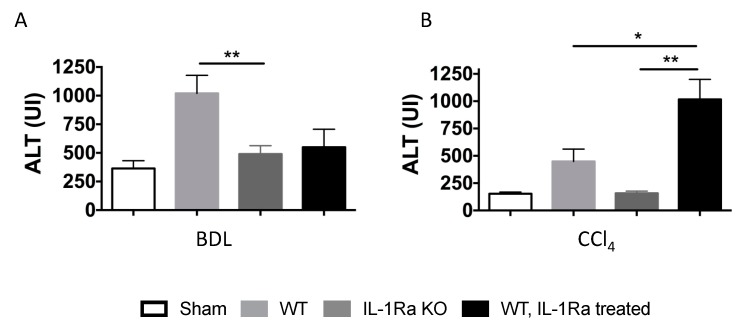
Serum alanine aminotransferase levels (ALT). Serum ALT levels were measured in sham mice and mice following 2–4 weeks BDL (**A**) or 6 weeks CCl-4 treatment (**B**). BDL and CCl-4 groups were respectively as follows: WT (*n* = 7 and *n* = 8), IL-1Ra KO (*n* = 8 and *n* = 5) or IL-1Ra treated mice (50 mg/kg/day) (*n* = 8 and *n* = 13). Sham treated mice were used as control for the BDL and CCl-4 groups (*n* = 8 and *n* = 6). * *p* < 0.05, ** *p* < 0.01, comparing two groups as indicated.

**Figure 7 ijms-20-01295-f007:**
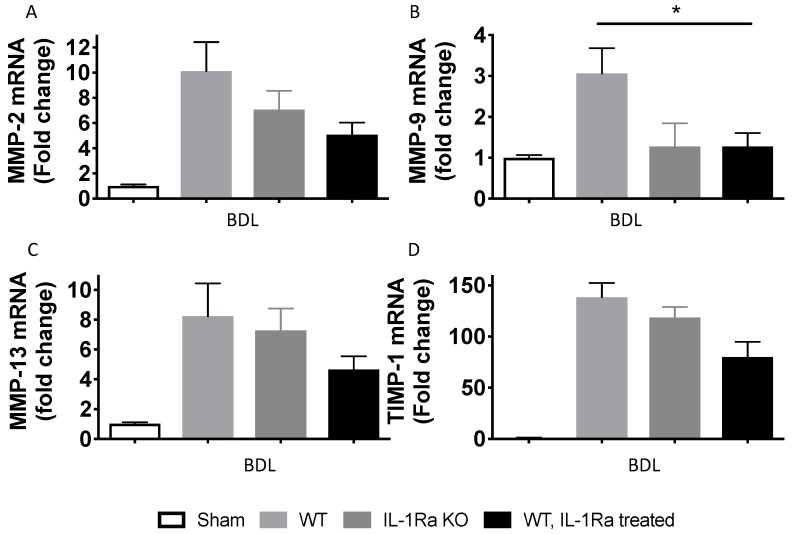
Matrix metalloproteinase (MMP) 2, 9 and 13 and tissue inhibitor of metalloproteinase 1 (TIMP-1) expression. MMP-2, 9 and 13 and TIMP-1 liver mRNA levels were measured by RT-PCR in sham mice and mice following 2–4 weeks BDL (**A**–**D**) or 6 weeks CCl-4 treatment (**E**–**H**). BDL and CCl-4 groups were respectively as follows: WT (*n* = 7 and *n* = 8), IL-1Ra KO (*n* = 7 and *n* = 4) or IL-1Ra treated mice (50 mg/kg/day) (*n* = 10 and *n* = 13). Sham treated mice were used as control for the BDL and CCl-4 groups (*n* = 8 and *n* = 4). * *p* < 0.05, ** *p* < 0.01, *** *p* < 0.001, comparing two groups as indicated.

**Table 1 ijms-20-01295-t001:** RT-PCR Primer Sequences.

Name	Primer-Probe Sequence 5′-3′	
	Forward	Reverse
**Mice**		
IL-1β	ACT CCT TAG TCC TCG GCC A	TGG TTT CTT GTG ACC CTG AGC
IL-1Ra	CTG CAC TTC CAC AGT CCA GA	ATA TGT GAT GCC CTG GTG GT
Collagen type I alpha 1	GCA TGG CCA AGA AGA CAT CC	CCT CGG GTT TCC ACG TCT C
MMP-2	TGG GGG AGA TTC TCA CTT TG	CAT CAC TGC GAC CAG TGT CT
MMP-9	AGT TGC CCC TAC TGG AAG GT	GTG GAT AGC TCG GTG GTG TT
MMP-13	AGT TGA CAG GCT CCG AGA AA	AGT TCG TTT GGG ACC ATT TG
TIMP-1	GCA TCT CTG GCA TCT GGC ATC	GAA GGC TGT CTG TGG GTG GG1 [8]
**Human**		
IL-1β	TCC AGG GAC AGG ATA TGG AG	TCT TTC AAC ACG CAG GAC AG [64]
IL-1Ra	AAG ACC AGT CCA TGA GGG AG	CTC CCC GAA AGA ACA TAA TCT C
α-SMA	CAT CTA TGA GGG CTA TGC CTT G	GTG AAG GAA TAG CCA CGC TC
MMP-9	TTG ACA GCG ACA AGA AGT GG	GCC ATT CAC GTC GTC CTT AT
MMP-2	Ref: QT00088396	Ref: QT00088396

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
