# Peer review of "Interleukin-1 Receptor Antagonist Modulates Liver Inflammation and Fibrosis in Mice in a Model-Dependent Manner"

_ijms, 2019, doi:10.3390/ijms20061295_

Round 1

Reviewer 1 Report

The study of Meier et al describes the effect of interleukin 1 receptor antagonist (IL-1Ra) in two models of liver damage. They find a clear effect on fibrotic development in this models, likely mediated by the effect of the molecule modulating the response to interleukin 1-beta. The authors provide a mechanism on this effect by an initial study on human hepatic stellate cells, where the effect of IL-1b is reversed by addition of recombinant IL-1Ra.

There are several methodological aspects of the study that should be considered.

Quantification of figures 1A and B. The observed effect of IL1beta effect on a-SMA content is stronger in the blot shown in Figure 1A than in the quantification of figure 1B. Could the authors comment on this discrepancy?

The quantification is a fold change measurement, but the authors do not indicate how are the values normalized, i.e., which is the 1 value.

The changes are measured at 48 hours. Do the authors have data on protein and/or gene expression at shorter incubation times?

The authors should test the effect of IL-1Ra on TGF-b induction of HSC in order to proof that the effect of IL-1Ra is specific to IL-1b induction.

In Figure 2, the title of the legend “IL-1β and IL-1Ra serum levels are upregulated in mice after bile duct ligation- or carbon tetrachloride-induced liver fibrosis.” does not seem to correspond with the data. The authors provide data on serum IL1beta and IL-1Ra in the BDL model (panels E and F), but not in the CCl4 model. Actually this is stated in the text (L113 not done in the CCl4 model). It is surprising that these data are lacking for the CCl4 model, as they are relevant for the distinction on the effect observed.

The presentation is confusing. The second bar should be labeled as WT, not BDL or CCl4.

IL1-Ra treated are WT animals treated with IL-1Ra?

Do the authors have data on IL-1RA KO mice treated with IL-1Ra? This is an important aspect to demonstrate that the recombinant protein can rescue the effect on the KO.

Why is IL-1Ra treatment not increasing the concentration of IL-1Ra in serum? Could the authors comment on this point?

Figure 4 and 5. Do the authors have a periportal-specific quantification? Or they just provide a general measurement of the tissue, assuming that HSC will be more abundant on the periportal zone?

Apart from Sup fig 1, the rest of the supplementary figures could be included in the main figures, as they are additional analysis of the same RNAs or aminotransferases, which are important biomarkers of the state of the model.

Discussion section. The authors propose that the effect of IL-1RA is mediated by its inhibitory effect on IL-1b. However, data on IL-1b KO mice treated with IL1-RA could provide essential clues on possible effects of IL-1Ra unrelated to IL-1b.

The authors propose an explanation of the bivalent effect of IL-1RA on the periportal specific effect of the BDL model, which implicates more directly the HSC on its response. Could the authors provide further data on the literature supporting this view? Are HSC not important on the fibrotic response in the CCl-4 model?

Minor points.

Line 71. Could the authors confirm that “IL1-KO knockout “is correct in this sentence? I thought they found a dual effect of IL-1Ra.

The authors should use abbreviations consistently. They use “CCl4” (L107, L113, L128) and “CCl-4”, “IL1-Ra” in figures 2, 3 instead of IL-1Ra.

L237 “…their fibrosis production” check grammar.

L332 IL-1Ra treatment was done, every day from BDL to sacrifice. What about the CCl4 treated mice of Figure 2D?

Author Response

Response to reviewers:

Reviewer 1

The study of Meier et al describes the effect of interleukin 1 receptor antagonist (IL-1Ra) in two models of liver damage. They find a clear effect on fibrotic development in this models, likely mediated by the effect of the molecule modulating the response to interleukin 1-beta. The authors provide a mechanism on this effect by an initial study on human hepatic stellate cells, where the effect of IL-1b is reversed by addition of recombinant IL-1Ra.

There are several methodological aspects of the study that should be considered.

Quantification of figures 1A and B. The observed effect of IL1beta effect on a-SMA content is stronger in the blot shown in Figure 1A than in the quantification of figure 1B. Could the authors comment on this discrepancy?

Response: Thank you very much for your comment. The GADPH quantification was in fact slightly lower in the IL-1beta treated groups. As a result, when normalized to GADPH, the aSMA expression is lowered but not as much as we would think when looking at the aSMA blot only. We redid the quantification of the western blots and we could confirm those results.

The quantification is a fold change measurement, but the authors do not indicate how are the values normalized, i.e., which is the 1 value.

Response: Thank you very much for your comment. We appreciate the reviewer’s question and for clarity, we recomputed all values and provided new graphs for all qPCR experiments using 1.0 as a reference value for controls (Figures 1-4, 7).

The changes are measured at 48 hours. Do the authors have data on protein and/or gene expression at shorter incubation times?

Response: We unfortunately did not perform the analysis at various time-points. Obtaining primary stellate cells from patients was challenging and the limited number of cells available did unfortunately not allow us to perform the analysis at different time-points.

The authors should test the effect of IL-1Ra on TGF-b induction of HSC in order to proof that the effect of IL-1Ra is specific to IL-1b induction.

Response: We agree with the reviewer that this is an interesting question. We did those experiments in immortalized human HSCs and found that IL-1Ra did not reverse the effect of TGF-b (see figure here below). This indeed suggests that IL-1Ra effect is specific to IL-1b induction. We added this observation in the manuscript (line 83).

In Figure 2, the title of the legend “IL-1β and IL-1Ra serum levels are upregulated in mice after bile duct ligation- or carbon tetrachloride-induced liver fibrosis.” does not seem to correspond with the data. The authors provide data on serum IL1beta and IL-1Ra in the BDL model (panels E and F), but not in the CCl4 model. Actually this is stated in the text (L113 not done in the CCl4 model). It is surprising that these data are lacking for the CCl4 model, as they are relevant for the distinction on the effect observed.

Response: We agree with the reviewer and changed the figure title for “IL-1β and IL-1Ra expression levels are upregulated in mice after BDL or CCl-4-induced liver fibrosis”. We did not repeat the quantification of IL-1Ra and IL-1b in the CCL-4 model. We unfortunately have no sera left from these mice to perform this analysis. However, given the similarity between the CCL-4 and BDL models on an expression level, and the correlation between gene expression and circulating protein levels within the BDL model, we can reasonably assume that, along with gene expression, IL-1Ra and IL-1b protein will be upregulated in WT mice in the CCL-4 model as well.

The presentation is confusing. The second bar should be labeled as WT, not BDL or CCl4.

Response: We agree with the reviewer and changed the label to “Sham, WT, IL-1Ra KO and WT, IL-1Ra treated" in all in vivo figures.

IL1-Ra treated are WT animals treated with IL-1Ra?

Response: Yes. For clarity, we changed the label to “Sham, WT, IL-1Ra KO and WT, IL-1Ra treated" in all in vivo figures.

Do the authors have data on IL-1RA KO mice treated with IL-1Ra? This is an important aspect to demonstrate that the recombinant protein can rescue the effect on the KO.

Response: We agree with the reviewer that this is an interesting question. We did those experiments in the BDL model and could demonstrate that IL-1Ra treatment could rescue IL-1Ra KO mice (see figure here below).

Why is IL-1Ra treatment not increasing the concentration of IL-1Ra in serum? Could the authors comment on this point?

Response: We agree with the reviewer and think that this is an important point. We previously discussed this point in detail in the discussion section, line 324: “Our primary hypothesis to explain this modest improvement [namely, the failure of IL1-Ra to change significantly the fibrosis levels compared to the WT group] is a failure of exogenous IL-1Ra to efficiently increase circulating levels due a possible negative feed-back mechanism regulating its own expression (Fig. 2F). Indeed, we observed that IL-1Ra blood levels were not much increased compared to the WT group despite daily anakinra administration. Another argument in favor of this hypothesis is that IL-1Ra expression by the liver was reduced upon IL-1Ra administration.”

Figure 4 and 5. Do the authors have a periportal-specific quantification? Or they just provide a general measurement of the tissue, assuming that HSC will be more abundant on the periportal zone?

Response: We provided a general measurement of the tissue.

Apart from Sup fig 1, the rest of the supplementary figures could be included in the main figures, as they are additional analysis of the same RNAs or aminotransferases, which are important biomarkers of the state of the model.

Response: Thank you for this comment. We agree and changed fig. suppl. 2 and 3 into Fig. 6 and 7.

Discussion section. The authors propose that the effect of IL-1RA is mediated by its inhibitory effect on IL-1b. However, data on IL-1b KO mice treated with IL1-RA could provide essential clues on possible effects of IL-1Ra unrelated to IL-1b.

Response: We agree with the reviewer that this point is important. We added this point to the discussion and modified that manuscript accordingly (line 331). Of note, in a confirmatory experiment (here above), we could demonstrate that IL-1Ra treatment can rescue IL-1Ra KO mice.

The authors propose an explanation of the bivalent effect of IL-1RA on the periportal specific effect of the BDL model, which implicates more directly the HSC on its response. Could the authors provide further data on the literature supporting this view? Are HSC not important on the fibrotic response in the CCl-4 model?

Response: We thank the reviewer for raising this important point. HSCs are known to be predominant in the pericentral area (Friedman, S. L., Hepatic stellate cells: protean, multifunctional, and enigmatic cells of the liver. Physiol Rev 2008, 88, (1), 125-72.) and Bronfenmajer S, et al. Fat-storing cells (lipocytes) in human liver. Arch Path 82: 447–453, 1966. Knowing that IL-1 did not promote HSC activation (in our in vitro experiments and shown by Pradere et al. Hepatology 2013), we hypothesized that when inflammation is mainly pericentral (CCL4 model), IL-1Ra absence (namely, more IL-1 signaling) ultimately partially block the profibrogenic signals. When inflammation is more periportal (BDL model), blocking inflammation at the pericentral area is counterproductive and leave the pericentral inflammation unopposed. We modified the text accordingly to further support our hypothesis (line 334). We agree that the exact cascade of signal between the cells implicated in the transition from inflammation to liver fibrosis largely remains to be investigated.   

Minor points.

Line 71. Could the authors confirm that “IL1-KO knockout “is correct in this sentence? I thought they found a dual effect of IL-1Ra.

Response: The sentence “IL-1Ra KO mice displayed increased fibrosis in the BDL model and reduced fibrosis in the CCl-4 model” seems correct and in accordance with our findings.

The authors should use abbreviations consistently. They use “CCl4” (L107, L113, L128) and “CCl-4”, “IL1-Ra” in figures 2, 3 instead of IL-1Ra.

Response: we thank the reviewer for this helpful comment. We corrected accordingly.

L237 “…their fibrosis production” check grammar.

Response: we thank the reviewer for this comment. We corrected accordingly.

L332 IL-1Ra treatment was done, every day from BDL to sacrifice. What about the CCl4 treated mice of Figure 2D?

Response: CCl-4 mice received daily IL-1Ra treatment as well. We added this information to the method section.

Reviewer 2 Report

In this study, Meier et al evaluated the effects of Interleukin-1 (IL-1) β signaling in human hepatic stellate cells (HSC) and the pathophysiological roles of IL-1β and IL-1 receptor antagonist (IL-1Ra) in mouse models of liver fibrosis induced by bile duct ligation (BDL) or carbon tetrachloride treatment (CCl4). Based on the results, the authors suggested that IL-1Ra has opposite effects in two liver fibrosis mouse models (IL-1Ra was detrimental in the CCL-4 model, whereas it was protective in the BDL model). In conclusion, the authors suggest that blocking IL-1-mediated inflammation with IL-1Ra may be beneficial only in selective liver fibrotic disease.

The manuscript is very well written and clear. This study will provide important information for the researchers in the study field.

The reviewer would like ask a question about the results of qPCR for IL-1Ra in mouse liver. As shown in Figure 2C and D, the IL-1Ra KO mouse groups showed the slight increase in IL-1Ra mRNA expression levels (particularly in CCl4 model). The reviewer confirmed that the primer pair for IL-1Ra can detect the Mus musculus interleukin 1 receptor antagonist using Primer-BLAST. I consider that the slight increase may not be significant, but it might be due to non-specific reaction, such as primer dimer. The authors should check the results again.

Author Response

Response to reviewers:

Reviewer 2

In this study, Meier et al evaluated the effects of Interleukin-1 (IL-1) β signaling in human hepatic stellate cells (HSC) and the pathophysiological roles of IL-1β and IL-1 receptor antagonist (IL-1Ra) in mouse models of liver fibrosis induced by bile duct ligation (BDL) or carbon tetrachloride treatment (CCl4). Based on the results, the authors suggested that IL-1Ra has opposite effects in two liver fibrosis mouse models (IL-1Ra was detrimental in the CCL-4 model, whereas it was protective in the BDL model). In conclusion, the authors suggest that blocking IL-1-mediated inflammation with IL-1Ra may be beneficial only in selective liver fibrotic disease.

The manuscript is very well written and clear. This study will provide important information for the researchers in the study field.

The reviewer would like ask a question about the results of qPCR for IL-1Ra in mouse liver. As shown in Figure 2C and D, the IL-1Ra KO mouse groups showed the slight increase in IL-1Ra mRNA expression levels (particularly in CCl4 model). The reviewer confirmed that the primer pair for IL-1Ra can detect the Mus musculus interleukin 1 receptor antagonist using Primer-BLAST. I consider that the slight increase may not be significant, but it might be due to non-specific reaction, such as primer dimer. The authors should check the results again.

Response: We agree with the reviewer and think that was worth checking. We checked the primers efficiency before using them and added a meltin curve at the end of the qPCR reaction. We could rule out the possibility that primers are not specific. We double checked again the Ct values and found that they were acceptable. The dissociation curves and the dissociation curves are clean and come out well as shown here below. Moreover, there are no second peaks in smaller sizes that would suggest the presence of primers dimers (see graph here below).

Reviewer 3 Report

The authors reported paradoxical effects of IL-1Ra on liver fibrosis in two different mouse models. Although the clear mechanisms behind their observations are still unclear, the study is still worth to be documented.

Minor questions

The manuscript is very well prepared except a few mistypings.

The authors used GeNorm method to analyze their RT-PCR, which is a little different from commonly used double delta Ct. The authors need to give some more details about the GeNorm method in their methods, especially how did they get the normalization factor. The fold change of their RT-PCR data is very confusing. What is the baseline of each genes? I also did not find any reference genes that were analyzed in this study.

Author Response

Response to reviewers:

Reviewer 3

The authors reported paradoxical effects of IL-1Ra on liver fibrosis in two different mouse models. Although the clear mechanisms behind their observations are still unclear, the study is still worth to be documented.

Minor questions

The manuscript is very well prepared except a few mistypings.

The authors used GeNorm method to analyze their RT-PCR, which is a little different from commonly used double delta Ct. The authors need to give some more details about the GeNorm method in their methods, especially how did they get the normalization factor. The fold change of their RT-PCR data is very confusing. What is the baseline of each genes? I also did not find any reference genes that were analyzed in this study.

Response: We thank the reviewer for this important comment. We added an additional explanation for the GeNorm method and provide a reference in the method section. Briefly, GeNorm was used to determine which normalization genes are the most stable in the sample population studied and select them for normalization. We computed a normalization factor for each individual sample represented by the geometric mean of the quantity values from the most stable normalization genes. Each sample will therefore have its own normalization factor. Unlike the Ct delta delta method, which uses only one gene and assumes that the PCR efficiency is 100% for all samples, our method therefore uses several standardization genes and considers the PCR efficiency individually for each sample. For more detail, please refer to the following paper that describe the GeNorm: Vandesompele J et al. Accurate normalization of real-time quantitative RT-PCR data by geometric averaging of multiple internal control, June 2002, Genome Biology 2002, 3(7).

Round 2

Reviewer 1 Report

The authors have improved the manuscript. It is incovenient that there were no samples to provide some of the additional data suggested. It seems importnt to include, at least as a supplementary table, the data on IL-1Ra rescue.

There are two points that could be explained by the authors.

My previous comment on "Why is IL-1Ra treatment not increasing the concentration of IL-1Ra in serum? Could the authors comment on this point?" The authors mention they explanation on the lack of effect on fibrosis. However, they observe an effect on a-SMA mRNA without increasing the concentration in serum. This was my point.

My comment "Line 71. Could the authors confirm that “IL1-KO knockout “is correct in this sentence? I thought they found a dual effect of IL-1Ra." Was referring to the last sentence:

"We thus highlight a dual and type-of-liver-injury-dependent effect of IL-1KO knockout." I do not think this sentence is correct.

Author Response

Response to reviewer:

Reviewer 1

The authors have improved the manuscript. It is inconvenient that there were no samples to provide some of the additional data suggested. It seems important to include, at least as a supplementary table, the data on IL-1Ra rescue.

Response: we agree with the reviewer’s comment. We added a supplementary figure showing the data on IL-1Ra rescue.

There are two points that could be explained by the authors.

My previous comment on "Why is IL-1Ra treatment not increasing the concentration of IL-1Ra in serum? Could the authors comment on this point?" The authors mention they explanation on the lack of effect on fibrosis. However, they observe an effect on a-SMA mRNA without increasing the concentration in serum. This was my point.

Response: we agree with the reviewer on the fact that IL-1Ra treatment has a mild effect on a-SMA mRNA levels. However, we believe that IL-1Ra given systemically is not fully equivalent to the endogenous natural section of IL-1Ra. An example of that discrepancy is better depicted by the difference observed in insulin levels between the different groups of mice after BDL. We measured serum insulin levels in all mice groups and observed a significant increase in insulin levels in mice after BDL in WT mice and KO mice (consistent with insulin resistance that is commonly observed after liver fibrosis/dysfunction (Garcia-Compean, D et al. Hepatogenous diabetes. Annals of hepatology 2009). However that increase was completely reversed by IL-1Ra treatment in WT mice (and KO mice) (see graph here below) consistently with the known anti-diabetic effect of exogenous IL-1Ra (Larsen, C. M. et al. Interleukin-1-receptor antagonist in type 2 diabetes mellitus. The New England journal of medicine 2007). This suggests that IL-1Ra treatment is not completely equivalent to IL-1Ra endogenous secretion regardless of its serum level. This might explain the effect on a-SMA mRNA levels in the apparent absence of significant further increase in IL-1Ra level in WT mice. We added these elements to the discussion of the manuscript.

My comment "Line 71. Could the authors confirm that “IL1-KO knockout “is correct in this sentence? I thought they found a dual effect of IL-1Ra." Was referring to the last sentence:

"We thus highlight a dual and type-of-liver-injury-dependent effect of IL-1KO knockout." I do not think this sentence is correct.

Response: we thank the reviewer for this helpful comment. We changed the sentence for

“We thus highlighted a dual effect of IL-1Ra that was dependent on the type of liver injury.”.